# Functionalized 10-Membered Aza- and Oxaenediynes through the Nicholas Reaction

**DOI:** 10.3390/molecules27186071

**Published:** 2022-09-17

**Authors:** Natalia A. Danilkina, Ekaterina A. Khmelevskaya, Anna G. Lyapunova, Alexander S. D’yachenko, Alexander S. Bunev, Rovshan E. Gasanov, Maxim A. Gureev, Irina A. Balova

**Affiliations:** 1Institute of Chemistry, Saint Petersburg State University (SPbU), Universitetskaya nab. 7/9, 199034 Saint Petersburg, Russia; 2Medicinal Chemistry Center, Tolyatti State University, 445020 Tolyatti, Russia; 3Center of Chemo- and Bioinformatics, I. M. Sechenov First Moscow State Medical University, 119991 Moscow, Russia; 4Department of Computational Biology, Sirius University of Science and Technology, 354349 Sochi, Russia

**Keywords:** alkynes, enediynes, heterocycles, benzo[*b*]thiophene, Bergman cyclization, Nicholas reaction, Sonogashira coupling, benzenesulfonamides

## Abstract

The scope and limitations of the Nicholas-type cyclization for the synthesis of 10-membered benzothiophene-fused heterocyclic enediynes with different functionalities were investigated. Although the Nicholas cyclization through oxygen could be carried out in the presence of an ester group, the final oxaenediyne was unstable under storage. Among the N-type Nicholas reactions, cyclization via an arenesulfonamide functional group followed by mild Co-deprotection was found to be the most promising, yielding 10-membered azaendiynes in high overall yields. By contrast, the Nicholas cyclization through the acylated nitrogen atom did not give the desired 10-membered cycle. It resulted in the formation of a pyrroline ring, whereas cyclization via an alkylated amino group resulted in a poor yield of the target 10-membered enediyne. The acylated 4-aminobenzenesulfonamide nucleophilic group was found to be the most convenient for the synthesis of functionalized 10-membered enediynes bearing a clickable function, such as a terminal triple bond. All the synthesized cyclic enediynes exhibited moderate activity against lung carcinoma NCI-H460 cells and had a minimal effect on lung epithelial-like WI-26 VA4 cells and are therefore promising compounds in the search for novel antitumor agents that can be converted into conjugates with tumor-targeting ligands.

## 1. Introduction

Enediyne antibiotics are an important class of natural products [1,2,3,4,5,6,7]. Although the first derivatives of 10-membered enediynes, Esperamicin [8] and Calicheamicin γ_1_^I^ [9,10], were isolated from bacteria almost forty years ago, interest in these unusual natural products and their use has continued [11,12] due to the unprecedented toxicity of these products to various cell lines [13]. The described biological activity of cyclic enediynes is associated with their ability to undergo the Bergman cyclization [14,15] with the formation of highly reactive diradicals that damage DNA and kill cancer cells [16] (Figure 1a). Currently, enediynes are the focus of the search for novel anticancer drugs based on the conjugation of natural enediyne warheads with monoclonal antibodies (MABs) to obtain antibody-drug conjugates (ADCs) [17,18,19]. Thus, two ADSs of Calicheamicin γ_1_^I^, Gemtuzumab ozogamicin (Mylotarg^®^) [20] and Inotuzumab ozogamicin (Besponsa^®^) [21], have been approved for the targeted treatment of various types of cancer (Figure 1b,c).

However, studies devoted to analogs of natural enediynes are also extremely important [22,23,24,25,26,27,28,29,30,31]. Recently, Nicolaou’s group reported a synthetic approach to tiancimycin B and uncialamycin analogs with promising antitumor activity (Figure 2a) [32]. Ding and Hu’s groups have successfully elaborated maleimide-assisted rearrangement and cycloaromatization (MARACA) of acyclic analogs of enediyne antibiotics that have been found to be active against different types of cancer cells [33,34,35,36] (Figure 2b).

Another recent focus in this field is the Bergman cyclization followed by the nucleophilic attack to 1,4-phenylene diradicals and the formation of halogen-substituted polyaromatic products [34,37] and the use of these products in organic synthesis (Figure 2c) [38]. Enolate ions as nucleophiles have recently been shown to be active in trapping 1,4-phenylene diradicals formed from 10-membered enediyne [39]. Moreover, intramolecular “nucleophilic trapping” of the diradicals formed by MARACA-type cyclization is possible [40].

Our efforts in this area have been concentrated on the intensive study of the molecular design of 10-membered heteroenediyne systems fused to heterocycles. Note that naturally occurring enediynes are protected from spontaneous Bergman cyclization due to specific structural features, and the enediyne core becomes reactive only after activation by special triggering [16]. Therefore, in the search for simple active cyclic analogs of natural enediynes lacking masking groups, the optimal balance between the stability of enediyne molecules during synthesis, isolation, and storage and the reactivity of these analogs in the Bergman cyclization is of decisive importance. We recently reported that this balance could be regulated by the nature of the fused heterocyclic core, as well as by the nature of the endocyclic σ-acceptor bound to the propargylic carbon atom of the enediyne system [41,42,43] (Figure 2d). Moreover, we proposed that the deviation of alkyne bond angles from 180° (alkyne bending) as well as the difference in bending of the ground state and the Bergman cyclization transition state can be used as parameters for the evaluation of the reactivity of annulated enediynes in the Bergman cyclization [41] instead of the «cd-distance» [44]. It is noteworthy that the same approach has recently been extended by Basak’s group to other families of enediyne [45].

Thus, the optimal balance between stability, the Bergman cyclization reactivity, and DNA cleavage ability was found for benzothiophene-fused azaenediynes [41,43], whereas O-enediynes were somewhat less stable but had slightly higher DNA damaging activity [42]. While searching for convenient synthetic methods for constructing strained 10-membered enediyne systems fused to a five-membered heterocycle, we found the Nicholas reaction to be the most promising and effective synthetic tool [42,43,46] because other reactions, such as ring-closing metathesis [47] and the Nozki–Hiama–Kishi reaction, do not work in this case [48].

The Nicholas reaction, namely, the alkylation of various nucleophilic functional groups with a stabilized Co_2_(CO)_6_-propargyl carbocation [49,50,51,52,53,54], has several characteristics necessary for the successful closing of a strained enediyne cycle. Thus, in acyclic precursors, the proximity of reaction centers contributes to the formation of a 10-membered ring; Co-protected enediynes are less strained and therefore more stable than their cobalt-free derivatives and can be stored for a long time as Co_2_(CO)_6_-enediyne complexes and simply can be deprotected from cobalt under mild conditions, if necessary.

However, all the heteroenediynes previously synthesized through the Nicholas-type cyclization lacked any functionality for further modification of the 10-membered core to control the solubility of target molecules, increase the affinity of these target molecules for cancer cells, and produce ADCs [41,42,43,46]. Here, we explored the possibilities of using Nicholas cyclization in the construction of functionalized aza- as well as oxaenediynes suitable for further synthetic modifications. We show that the Nicholas reaction though O- and N-atoms, can be used to synthesize benzothiophene-fused enediynes with different functional groups (Figure 2e): ester, *o*-nosyl functional groups, or a terminal triple bond. Arenesulfonamide nucleophilic group was found to be the optimal moiety for the synthesis of functionalized enediynes.

## 2. Results

Searching for the optimal nucleophilic functional group for the synthesis of functionalized 10-membered oxa- and azaenediynes, we chose the target structures **I–V** (Figure 3), which can be obtained using cyclization through OH (**I**), NHSO_2_Ar (**II, III**), NHBn (**IV**) and NHBz (**V**). 

Synthesis of all the target structures is based on a combination of electrophile-promoted cyclization of the starting diacetylene and the subsequent Sonogashira coupling to construct unsymmetrically substituted acyclic enediynes with the required functionalities at both triple bonds followed by the regioselective formation of Co_2_(CO)_6_-complexes for the further Nicholas cyclization [42,46,55] (Figure 3). The key intermediate compound for all the structures was 3-iodo-2-(3-methoxyprop-1-yn-1-yl)benzo[*b*]thiophene (**1**), which is available synthetically at the multigram scale as has been previously reported [46].

### 2.1. Synthesis of Oxaenediyne I

A four-step synthesis of the ester-functionalized O-enediyne **I** was started from 3-iodobenzothiophene **1**. Desilylation of the functionalized alkyne **2** and the Sonogashira coupling were carried out in one pot using KF/MeOH/DMF as the desilylation source (Figure 1) [56]. Further complexation of the enediyne **3** with Co_2_(CO)_8_ proceeded regioselectively with the formation of the cobalt complex **4** at the C2-triple bond. The higher selectivity for the C2-triple bond compared with the nonfunctionalized enediyne can be explained by the higher steric hindrance of the triple bond at the C3 position [46].

The Nicholas reaction of the ester-functionalized complex **4** proceeded under optimized conditions (1.5 equiv. of FB_3_·Et_2_O) to afford the cyclic product **5** in good yield. We recently showed that using tetrabutylammonium fluoride (TBAF) hydrate in an aqueous acetone solution increases the yield of cobalt-free 10-membered enediynes in the decomplexation step [43]. Therefore, we applied these conditions to the ester-functionalized Co-complex **5,** as well as to the previously reported Co-complex of the nonfunctionalized O-enediyne **6**, which allowed us to obtain the enediyne **I** and noticeably increase the yield of the enediyne **7** at the decomplexation step compared with previous results [46]. However, the target ester-containing enediyne **I** was significantly less stable than its unsubstituted analog **7** and gave traces of the Bergman cyclization product in experiments with NMR detection. Then, the Bergmann cyclization of the enediynes **I** and **7** was carried out in *i*-PrOH at 45–50 °C, and both cyclization products **8** and **9** were isolated in high yields.

### 2.2. Synthesis of Azaenediynes II–V

We recently showed that the amino functional group protected by the tosyl group (NH-Ts) is very efficient for the synthesis of azaenediyne systems through the Nicholas cyclization, and there was an optimal balance between the stability and DNA-damaging activity of the resulting N-Ts-enediyne [42]. Therefore, we decided to use an arenesulfonamide fragment to introduce functional groups into the N-enediyne molecule. Two types of functionalized arenesulfonamide moieties were used: 2-nosyl as an easily removable protecting group and a sulfanilamide moiety with an NH_2_ group acylated with hex-5-ynoic acid.

The N-Ns (2-nosyl, 2-nitrobenzenesulfonyl) enediyne **III** was synthesized similarly to the N-Ts enediyne [42] (Figure 2).

Thus, the starting Co-complex **12a** for the Nicholas reaction was obtained without any difficulties and in high yield. However, the Nicholas reaction of the NH-Ns group formed the product **13a** in a considerably lower yield (46%) compared with the NH-Ts function (76%) and required a higher quantity of a Lewis acid (8 equiv. instead of 1.5 equiv.) [42], which can be explained both by the steric hindrance and lower nucleophilicity of NH-Ns.

To synthesize the *N*-hex-5-ynoyl enediyne **III**, the obtained *p*-NO_2_Ph-substituted enediyne **11b** was reduced by Zn in the AcOH/DCM system to the NH_2_-derivative **11c**, which was then converted to the Co-complex **12b**. Then, the Co-complex **12b** was acylated with hex-5-ynoyl chloride to produce the nontrivial triacetylenic compound **12c**, in which only one triple bond out of three was converted to the Co complex. The Nicholas cyclization of the N-protected/functionalized Co-complex **12c** proceeded smoothly to give the desired cyclic Co-complex **13b** in good yield (64%). It should be emphasized that the Nicholas cyclization of the Co-complex with the free NH_2_ group **12b** did not proceed at all. Therefore, protection, along with functionalization at the stage of an acyclic Co-complex, is a necessary synthetic step for producing a 10-membered enediyne with a terminal triple bond. Finally, we investigated the last decomplexation step, which proceeded in good to high yields to give the N-enediynes **II** and **III,** which were stable under isolation and storage. We also tested decomplexation in aqueous acetone for the Co-complex of the N-Ts-enediyne **13c**, which has been previously reported [42]. In this case, we succeeded in increasing the yield of **14** at the decomplexation step from 45% (in anhydrous acetone) [42] to 88% (in aqueous acetone).

It is known that the Nicholas-type cyclization can proceed using amide functional groups [57] and even through secondary amino groups in the presence of DIPEA [58]. Therefore, we decided to test these functional groups, which could also be useful for the functionalization of enediynes. Therefore, cyclization using NHBn and NHBz groups was also studied (Figure 3).

The corresponding starting materials, the Co-complexes **17a** and **17b**, were synthesized without any difficulties starting from iodobenzothiophene **1** and the corresponding functionalized terminal alkynes **15a** and **15b** in two steps (Figure 3); however, cyclization failed in both cases. Thus, the conditions tested for the Nicholas cyclization of the NHBn Co-complex **17a** (BF_3_·Et_2_O; HBF_4_·Et_2_O and HBF_4_·Et_2_O/DIPEA) gave the desired cyclic compound **18** in low yields due to the complexation of the NHBn group with the acid. Even generation of the carbocation with HBF_4_·OEt_2_ followed by deprotonation of the [NH_2_Bn]^+^ group with DIPEA only gave a 12% yield of the cyclic enediyne **18**. Therefore, the basicity of the secondary amino function should be considered a strong limitation of the Nicholas reaction in the case of enediyne systems.

Cyclization of the complex **17b** using an NH-benzoyl moiety as a nucleophilic group did not give the desired 10-membered enediyne at all (Figure 3). The main product of the reaction was the pyrroline derivative **19** due to the electrophile-promoted cyclization of the NHBz functional group at the free triple bond. This result can be explained by the higher steric hindrance of the planar NHBz group compared with that of the tetrahedral arenesulfonamide functional group. Thus, we have proven that the sulfonamide moiety remains the functional group of choice when using the aza-Nicholas reaction to synthesize 10-membered N-enediynes.

### 2.3. Biological Activity of Cyclic Enediynes

All the synthesized 10-membered enediynes (**I–III, 7,** and **14**) were tested for their effect on the growth of NCI-H460 lung carcinoma and WI-26 VA4 lung epithelial-like cell lines using the MTT colorimetric test [59,60] with the cytotoxic drug etoposide as a positive control. All the enediynes at a concentration of 75 μM displayed moderate cytotoxicity toward cancer cells and had less effect on normal fibroblasts (Figure 4).

These data correspond with the previously estimated ability of benzothiophene-fused enediynes to cleave plasmid DNA [41,42,46]. Therefore, the observed cytotoxic activity of enediynes is assumed to be associated with their DNA damaging effect. However, it is clear that for DNA to be affected and destroyed in cells, a molecule must have a sufficient hydrophilic-lipophilic balance and the ability to penetrate into cells and avoid various drug resistance mechanisms. Therefore, to improve the cytotoxic effect of enediynes, the functional design of benzothiophene-fused enediyne molecules should be elaborated. From this point of view, considering the absence of significant differences in cytotoxicity, N-enediynes are the most promising compounds for further development of antitumor agents. Thus, functionalized derivatives of N-enediynes are stable and synthetically accessible and can be used for further conjugation with ligands with an affinity for cancer cells.

## 3. Discussion

The scope and limitation of the Nicholas-type cyclization for the synthesis of various 10-membered azaenediynes, as well as oxa-analog were studied. We used two types of heteroatoms–xygen and nitrogen, to choose which type of heteroenediynes and which type of nucleophilic functional groups are most suitable for the synthesis of heteroenediynes that have additional functionality for further modification with tumor-targeting ligands.

We showed that functionalized O-enediyne with an ester group attached to the enediyne core is synthetically accessible through the O-Nicholas reaction. However, further functionalization is limited because of the low stability of O-enediynes.

N-Nicholas cyclization through three types of N-containing nucleophilic groups–amino, amido, and arenesulfonamido, was studied. We proved that an arenesulfonamide fragment is the optimal functional group to realize a high-yielded synthesis of 10-membered azaenediynes. Moreover, this group can serve as a site for the introduction of additional functional groups for further modification of cyclic enediynes using click chemistry. For this purpose, the 4-aminobenezenesulfonamide moiety should be acylated with acid derivatives containing a functional group tolerant to the Nicholas cyclization conditions. We demonstrated that this strategy could be applied to the synthesis of azaenediyne with a free terminal triple bond in the arenesulfonamide linker part.

While amino and amido nucleophilic groups also offer great potential as linkers for attaching clickable groups to an enediyne core, neither the secondary amino group nor the amido functional group is suitable for the closure of the 10-membered azaendiyne, which is a limitation of the aza-Nicholas cyclization.

All the synthesized cyclic enediynes were tested as potential anticancer compounds and showed moderate activity against NCI-H460 lung carcinoma and had a minimal effect on WI-26 VA4 lung epithelial-like cells. Thus, the modification of azaenediynes through the 4-aminobenezenesulfonamide moiety can be used in the future to synthesize enediyne conjugates with higher antitumor efficacy.

## 4. Materials and Methods

### 4.1. General Information and Methods

Solvents, reagents, and chemicals used for reactions were purchased from commercial suppliers. The chemicals were used without further purification. Catalyst Pd(PPh_3_)_4_ and Co_2_(CO)_8_ were purchased from Sigma-Aldrich. 3-Iodo-2-(3-methoxyprop-1-yn-1-yl)benzo[*b*]thiophene (**1**) [46], methyl-3-hydroxy-6-(trimethylsilyl)hex-5-ynoate (**2**) [61], *N*-(but-3-yn-1-yl)-2-nitrobenzenesulfonamide (**10a**) [62], *N*-benzylbut-3-yn-1-amine (**15a**) [63], *N*-(but-3-yn-1-yl)benzamide (**15b**) [64], Co-complexes **6 [46]** and **13c [42]** were synthesized according to known procedures without any modifications.

Solvents were dried under standard conditions. Purification and drying of DCM were carried out in accordance with the literature procedure using CaH_2_ [65]. The Sonogashira coupling, the synthesis of Co-complexes, the Nicholas cyclization, and the Bergman cyclization were carried out under argon in oven-dried glassware. Other reactions were carried out under air unless stated otherwise. Evaporation of solvents and concentration of reaction mixtures were performed under vacuum at 20 °C (for the enediynes **I–III, 7, 14**) and 35 °C (for other compounds) on a rotary evaporator. TLC was carried out on silica gel plates (Silica gel 60, UV 254) with detection by UV or staining with a basic aqueous solution of KMnO_4_. A normal-phase silica gel (Silica gel 60, 230−400 mesh) was used for preparative column chromatography. ^1^H and ^13^C{^1^H} and DEPT NMR spectra were recorded at 400 (or 500) and 101 (or 125) MHz, respectively, at 25 °C in CDCl_3_, acetone-*d_6,_* or CD_3_CN without an internal standard. The ^1^H NMR data are reported as chemical shifts (δ), multiplicity (s, singlet; d, doublet; t, triplet; q, quartet; m, multiplet; br, broad), coupling constants (*J*, given in Hz), and number of protons. The ^13^C{^1^H} NMR data are reported as chemical shifts (δ). Chemical shifts for ^1^H and ^13^C are reported as δ values (ppm) and referenced to residual solvents (δ = 7.26 ppm for ^1^H; δ = 77.16 ppm for ^13^C for spectra recorded in CDCl_3_ and δ = 2.05 ppm for ^1^H; δ = 29.84 ppm for ^13^C for spectra recorded in acetone-*d_6_* and δ = 1.94 ppm for ^1^H; δ = 1.32 ppm for ^13^C for spectra in CD_3_CN). For copies of NMR spectra of all new compounds see the Appendix A. High-resolution mass spectra were determined for solutions of all compounds in MeOH using ESI in the mode of positive ion registration with a TOF mass analyzer. For copies of ESI HRMS spectra of key products **I–III, 18, 19** see the Appendix A.

### 4.2. Experimental Details: General Procedures

#### 4.2.1. General Procedure (A) for the Synthesis of Acyclic Enediynes **11a,b**; **16a,b**

To a stirred, degassed solution of 3-iodo-2-(3-methoxyprop-1-yn-1-yl)benzo[*b*]thiophene (**1**) (1.00 equiv.) in anhydrous dimethylformamide (DMF) in a vial or a Schlenk flask were added alkyne (1.05−2.00 equiv.), KF (5.00–8.00 equiv.), Pd(PPh_3_)_4_ (5 mol%), and CuI (15 mol%) under atmosphere of an Ar. The reaction vessel was sealed, degassed, and flushed with Ar. The reaction mixture was stirred at 40–60 °C for the corresponding time (TLC monitoring). After completion of the reaction, the reaction mixture was cooled, poured into a saturated aqueous solution of NH_4_Cl, and extracted with ethyl acetate. The combined organic layers were washed two times with brine and dried over anhydrous Na_2_SO_4_. The solvent was evaporated under reduced pressure, and the residue was purified by column chromatography.

#### 4.2.2. General Procedure (B) for the Synthesis of Acyclic Enediyne Co_2_(CO)_6_ Complexes **4**; **12a,b**; **17a,b**

To a 0.005 M solution of acyclic enediyne (1.00 equiv.) in anhydrous toluene was added octacarbonyl dicobalt (1.05–1.15 equiv.), and the mixture was stirred under argon at room temperature for the corresponding time. The solvent was evaporated under reduced pressure, and the residue was purified by column chromatography.

#### 4.2.3. General Procedure (C) for the Synthesis of Cyclic Co_2_(CO)_6_-Complexes **5**; **13a,b**; **18** by the Nicholas Reaction and the Synthesis of Pyrroline Derivative **19**

To an argon-flushed, cooled (0°C) stirred solution of Co_2_(CO)_6_-complex of an acyclic enediyne (1.00 equiv.) in anhydrous DCM (c = 0.001 M) was added boron trifluoride diethyl etherate (1.50–8.00 equiv.). The resulting mixture was allowed to warm to room temperature and was stirred at room temperature until the reaction was complete (TLC). Then the reaction mixture was quenched with a saturated aqueous solution of NaHCO_3_. The organic layer was separated, washed with brine, dried over anhydrous Na_2_SO_4_, and concentrated under reduced pressure to yield a crude product, which was purified by column chromatography.

#### 4.2.4. General Procedure (D) for the Synthesis of 10-Mebered Enediynes I–III, **7**, **14** by the Deprotection of Acyclic Co_2_(CO)_6_-complexes from Cobalt

To a stirred solution of cyclic Co_2_(CO)_6_ complex (1.00 equiv., c = 0.006 M) in a mixture of acetone/water (15:1, *v/v*), tetrabutylammonium fluoride (TBAF) hydrate (calculated for TBAF × H_2_O) or trihydrate (calculated for TBAF × 3H_2_O), was added in several portions until the starting Co-complex was consumed, as indicated by TLC. The total amount of TBAF hydrate or TBAF·trihydrate varied from 23.5 to 65.8 equiv. After completion of the reaction, the reaction mixture was filtered through a pad of Celite using fritted filter funnel, the sorbent was washed with acetone, and the resulting solution was concentrated under reduced pressure at 20°C to ~ 1/5 of the original volume; the resulting mixture was mixed with ethyl acetate and brine. The organic layer was separated, and the aqueous layer was extracted with ethyl acetate. The combined organic layers were washed three times with brine, dried over anhydrous Na_2_SO_4_, and the solvent was evaporated under reduced pressure at 20°C. The crude product was purified by column chromatography.

### 4.3. Experimental Details: Specific Procedures

**Methyl 3-hydroxy-6-(2-(3-methoxyprop-1-yn-1-yl)benzo[*b*]thiophen-3-yl)hex-5-ynoate (3).** To a stirred solution of 3-iodobenzothiophene **1** (391 mg, 1.19 mmol, 1.00 equiv.) in DMF (8.00 mL) in a vial were added methyl-3-hydroxy-6-(trimethylsilyl)hex-5-ynoate (**2**) (383 mg, 1.79 mmol, 1.50 equiv.), KF (346 mg, 5.96 mmol, 5.00 equiv.), Pd(PPh_3_)_4_ (68.8 mg, 0.0600 mmol, 5 mol%). The reaction vial was evacuated and flushed with Ar several times. After that, CuI (34 mg, 0.179 mmol, 15 mol%) was added, and the reaction vial was sealed, evacuated, and flushed with Ar. Then, MeOH (382 mg, 11.9 mmol, 0.482 mL, 10.0 equiv.) was added with a syringe, and the reaction mixture was stirred at 40 °C for 13 h (TLC control). The reaction mixture was cooled, poured into a saturated aqueous solution of NH_4_Cl (150 mL), and extracted with ethyl acetate (3 × 100 mL). The combined organic layers were washed with a saturated solution of NH_4_Cl (200 mL) and two times with brine (2 × 200 mL), dried over anhydrous Na_2_SO_4_, and concentrated under reduced pressure to give the crude product, which was purified by column chromatography on silica using hexane/ethyl acetate (2:1) as the eluent to give enediyne **3** (400 mg, 98%) as a yellow solid. ^1^H NMR (400 MHz, CDCl_3,_ δ): 7.86–7.80 (m, 1H), 7.75–7.69 (m, 1H), 7.44–7.38 (m, 2H), 4.43 (s, 2H), 4.38–4.32 (m, 1H), 3.74 (s, 3H), 3.48 (s, 3H), 2.90–2.69 (m, 5H). ^13^C{^1^H}NMR (101 MHz, CDCl_3,_ δ): 172.9, 138.7, 138.6, 126.4, 125.5, 125.3, 123.5, 123.4, 122.3, 95.0, 92.6, 79.6, 76.6, 66.9, 60.7, 58.0, 52.0, 40.2, 27.9. HRMS (ESI) *m*/*z*: [M+Na]^+^ Calcd for C_19_H_18_O_4_SNa^+^, 365.0804; Found, 365.0818.

**Hexacarbonyl (methyl (3-hydroxy-6-(2-(3-methoxyprop-1-(1,2-η^2^)-yn-1-yl)benzo[*b*]thiophen-3-yl)hex-5-ynoate)dicobalt (4).** Co-complex **4** was synthesized from enediyne **3** (209 mg, 0.610 mmol, 1.00 equiv.) and octa-carbonyl dicobalt (230 mg, 0.670 mmol, 1.10 equiv.) in absolute toluene (122 mL, *c* = 0.005M) in accordance with the General Procedure (B). The reaction time was 1 h. Purification of the crude product by column chromatography using hexan/ethyl acetate (2:1) as the eluent gave Co-complex **4** (326 mg, 85%) as a dark red solid. ^1^H NMR (400 MHz, Acetone-*d_6_*, δ): 7.91–7.88 (m, 2H), 7.49–7.44 (m, 2H), 4.99 (s, 2H), 4.46–4.38 (m, 2H: 1H–OH; 1H–CH), 3.65 (s, 3H), 3.61 (s, 3H), 2.87 (d, *J* = 5.8 Hz, 2H), 2.64 (dd, *J* = 15.5, 7.9 Hz, 1H), signal from one H atom overlaps with water signal at 2.81 ppm. ^13^C{^1^H}NMR (101 MHz, Acetone-*d_6_*, δ): 200.1, 172.4, 146.1, 141.8, 139.5, 126.8, 126.3, 123.9, 123.2, 118.8, 98.4, 97.8, 78.1, 77.2, 74.0, 67.4, 59.2, 51.7, 41.9, 29.2. HRMS (ESI) *m*/*z*: [M+Na]^+^ Calcd for C_25_H_18_Co_2_O_10_SNa^+^, 650.9177; Found, 650.9183.

**Hexacarbonyl****(methyl ((2-((1,2-η^2^)-1,2,7,8-tetradehydro-5,6-dihydro-3*H*-benzo[4,5]thieno[2,3-*e*]oxecin-5-yl))acetate)dicobalt (5).** Compound **5** was synthesized from complex **4** (139 mg, 0.221 mmol, 1.00 equiv.) and boron trifluoride–diethyl ether (47.1 mg, 41.0 mmL, 0.331 mmol, 1.50 equiv.) in absolute DCM (222 mL, *c* = 0.001M) in accordance with the General Procedure (C); the reaction time was 1 h. Purification of the crude product by column chromatography using hexane/ethyl acetate (5:1) as the eluent gave unconverted complex **4** (26.7 mg, 19%) and cyclic Co-complex **5** (66.8 mg, 51%) as a dark red oil. ^1^H NMR (400 MHz, Acetone-*d_6_*, δ): 7.95–7.91 (m, 1H), 7.80–7.76 (m, 1H), 7.50–7.43 (m, 2H), 5.29 (d, *J* = 12.1 Hz, 1H), 5.20 (d, *J* = 12.1 Hz, 1H), 4.59–4.52 (m, 1H), 3.71 (s, 3H), 2.73–2.68 (m, 2H), signals from two H atom overlap with water signal at 2.79 ppm. ^13^C NMR (125 MHz, Acetone-*d_6_*, δ): 200.0, 171.9, 150.4, 140.0, 139.4, 126.7, 126.5, 123.7, 123.6, 117.6, 99.8, 97.1, 80.7, 79.62, 79.58, 75.0, 51.9, 41.5, 27.5. HRMS (ESI) *m*/*z*: [M+H]^+^ Calcd for C_24_H_15_Co_2_O_9_S^+^, 596.9095; Found, 596.9106. 

**Methyl 2-(1,2,7,8-tetradehydro-5,6-dihydro-3*H*-benzo[4,5]thieno[2,3-*e*]oxecin-5-yl)acetate (I).** The decomplexation from cobalt for complex **5** (66.0 mg, 0.111 mmol, 1.00 equiv.) was carried out in accordance with the General Procedure for the Co-complexes deprotection (D) using TBAF trihydrate (820.7 mg, 2.6 mmol, 23.5 equiv.) in an acetone/water mixture (15:1, *v/v*, 18.3 mL, *c* = 0.006 M). TBAF trihydrate was added in eight portions with an interval of 0.5 h. The reaction time was 3.5 h. The crude product was purified by column chromatography using hexane/acetone (30:1) as the eluent to give enediyne **I** (23.2 mg, 68%) as a light yellow solid. ^1^H NMR (400 MHz, Acetone-*d_6_*, δ): 7.97–7.95 (m, 1H), 7.79–7.76 (m, 1H), 7.51–7.45 (m, 2H), 4.67 (d, *J* = 17.7 Hz, 1H), 4.56 (d, *J* = 17.7 Hz, 1H), 4.48–4.42 (m, 1H), 3.69 (s, 3H), 2.76–2.69 (m, 4H). ^13^C{^1^H} NMR cannot be measured due to the instability of the enediyne **I** under the measurements. HRMS (ESI) *m*/*z*: [M+Na]^+^ Calcd for C_18_H_14_O_3_SNa^+^ 333.0556; Found, 333.0562.

**1,2,7,8-Tetradehydro-5,6-dihydro-3*H*-benzo****[4,5]thieno[2,3-*e*]oxecine (7).** Enediyne **7** was synthesized in accordance with the General Procedure (D) for the Co-complexes deprotection from Co-complex **6** (78.6 mg, 0.150 mmol, 1.00 equiv.) and TBAF hydrate (1.88 g, 6.73 mmol, 45.0 equiv.) in an acetone/water mixture (15:1, *v/v*, 25.0 mL, *c* = 0.006 M). TBAF hydrate was added in six portions with an interval of 0.5 h. The reaction time was 2.5 h. The crude product was purified by column chromatography using hexane/ethyl acetate (15:1) as the eluent to give O-enediyne **7** (22.7 mg, 64%) as **a** light yellow solid. ^1^H NMR (400 MHz, CDCl_3_) δ 7.79–7.75 (m, 2H), 7.44–7.35 (m, 2H), 4.48 (s, 2H), 4.17 (d, *J* = 4.8 Hz, 2H), 2.74 (d, *J* = 4.8 Hz, 2H). ^1^H NMR spectrum corresponds with the data reported earlier [46].

**Methyl 2-(3,4-dihydro-1*H*-benzo[4,5]thieno[3,2-*g*]isochromen-3-yl)acetate (8).** The solution of enediyne I (10.0 mg, 0.0322 mmol, 1.00 equiv.) and isopropanol (32.2 mL, c = 0.001 M) in a sealed vial were degassed accurately and flashed with Ar. Then the reaction mixture was stirred at 45 °C for 14 h. The solvent was evaporated under reduced pressure, and the crude product was purified by column chromatography using hexane/acetone (5:1) as the eluent to give the Bergman cyclization product **8** (8.4 mg, 83%) as a yellowish solid. ^1^H NMR (400 MHz, CDCl_3_) δ 8.11–8.07 (m, 1H), 7.88 (s, 1H), 7.84–7.80 (m, 1H), 7.49 (s, 1H), 7.46–7.41 (m, 2H), 4.99 (s, 2H), 4.28–4.21 (m, 1H), 3.76 (s, 3H), 3.05–2.94 (m, 2H), 2.78 (dd, *J* = 15.4, 7.8 Hz, 1H), 2.66 (dd, *J* = 15.4, 5.1 Hz, 1H). ^13^C{^1^H} NMR (100 MHz, CDCl_3_) δ 171.7, 139.7, 137.6, 135.3, 134.6, 133.7, 129.5, 126.8, 124.5, 123.0, 121.6, 121.5, 118.3, 71.9, 68.7, 52.04, 41.0, 34.0. HRMS (ESI) *m*/*z*: [M+H]^+^ Calcd for C_18_H_17_O_3_S, 313.0893; Found, 313.0883.

**3,4-dihydro-1*H*-benzo[4,5-*b*]thieno[3,2-*g*]isochromene (9).** The solution of enediyne **7** (5.30 mg, 0.0222 mmol) in isopropanol (22.2 mL, c = 0.001 M) in a sealed vial was degassed accurately and flashed with Ar. Then the reaction mixture was stirred at 50 °C for 12 h. The solvent was evaporated under reduced pressure, and the crude product was purified by column chromatography using hexane/ethyl acetate (5:1) as the eluent to give the product of the Bergman cyclization **9** (5.3 mg, 100%) as a white solid. ^1^H NMR (400 MHz, CDCl_3_) δ 8.12–8.09 (m, 1H), 7.91 (s, 1H), 7.84–7.80 (m, 1H), 7.47 (s, 1H), 7.45–7.41 (m, 2H), 4.93 (s, 2H), 4.06 (t, *J* = 5.7 Hz, 2H), 3.07 (t, *J* = 5.7 Hz, 2H). ^13^C{^1^H} NMR (101 MHz, CDCl_3_) δ 139.7, 137.4, 135.4, 134.5, 134.4, 130.1, 126.7, 124.5, 123.0, 121.8, 121.5, 118.4, 68.6, 65.8, 28.8. HRMS (ESI) *m*/*z*: [M+Na]^+^ Calcd for C_15_H_12_OSNa, 263.0508; Found, 263.0501.

***N*-(But-3-yn-1-yl)-4-nitrobenzenesulfonamide (10b).** Compound **10b** was synthesized in accordance with the procedure for the corresponding NHTs analogue[66] from *tert*-butyl but-3-yn-1-yl((4-nitrophenyl)sulfonyl)carbamate (2.53 g, 7.14 mmol, 1.0 equiv.) and trifluoroacetic acid (12.2 g, 7.97 mL, 0.107 mol, 15.0 equiv.). Yield 65% (1.19 g). ^1^H NMR (400 MHz, Acetone-*d_6_*) δ 8.51–8.36 (m, 2H), 8.26–7.91 (m, 2H), 7.06 (t, *J* = 5.5 Hz, 1H), 3.20–3.15 (m, 2H), 2.43–2.37 (m, 2H–CH_2_, 1H–C≡CH). ^1^H NMR spectrum corresponds with the data reported earlier [67].

***N*-(4-(2-(3-methoxyprop-1-yn-1-yl)benzo[*b*]thiophen-3-yl)but-3-yn-1-yl)-2-nitrobenzenesulfonamide (11a).** Enediyne **11a** was synthesized in accordance with the General Procedure (A) for the Sonogashira coupling from 3-iodobenzothiophene **1** (100 mg, 0.303 mmol, 1.00 equiv.), *N*-(but-3-yn-1-yl)-2-nitrobenzenesulfonamide (**10a**) (100 mg, 0.394 mmol, 1.3 equiv.), KF (88.0 mg, 1.52 mmol, 5.00 equiv.), Pd(PPh_3_)_4_ (17.5 mg, 0.015 mmol, 5 mol%) and CuI (8.66 mg, 0.046 mmol, 15 mol%) in anhydrous DMF (5.00 mL) at 50 °C. The reaction time was 5 h. The crude product was purified by column chromatography using hexane/ethyl acetate (2:1) as the eluent to give enediyne **11a** (130 mg, 94%) as a yellow oil. ^1^H NMR (400 MHz, CDCl_3,_ δ): 8.15 (dd, *J* = 7.7, 1.4 Hz, 1H), 7.73–7.69 (m, 3H), 7.65–7.55 (m, 2H), 7.42–7.38 (m, 2H), 5.83 (t, *J* = 6.0 Hz, 1H), 4.43 (s, 2H), 3.53–3.41 two signals overlap (m, 2H, s, 3H), 2.81 (t, *J* = 6.5 Hz, 2H). ^13^C{^1^H} NMR (100 MHz, CDCl_3,_ δ): 148.0, 138.6, 138.4, 134.2, 133.5, 132.9, 130.8, 126.5, 125.8, 125.5, 125.3, 123.4, 122.7, 122.2, 95.4, 92.6, 79.2, 76.1, 60.7, 58.0, 42.9, 21.5. HRMS (ESI) *m*/*z*: [M+Na]^+^ Calcd for C_22_H_18_N_2_O_5_S_2_Na^+^, 477.0549; Found, 477.0551.

***N*-(4-(2-(3-methoxyprop-1-yn-1-yl)benzo[*b*]thiophen-3-yl)but-3-yn-1-yl)-4-nitrobenzenesulfonamide (11b).** Enediyne **11b** was synthesized in accordance with the General Procedure (A) for the Sonogashira coupling from 3-iodobenzothiophene **1** (1.07 g, 3.26 mmol, 1.00 equiv.), *N*-(but-3-yn-1-yl)-4-nitrobenzenesulfonamide (**10b**) (870 mg, 3.42 mmol, 1.05 equiv.), KF (1.52 g, 26.1 mmol, 8.00 equiv.), Pd(PPh_3_)_4_ (188 mg, 0.163 mmol, 5 mol%) and CuI (93.0 mg, 489 µmol, 15 mol%) in DMF (25.0 mL) at 60 °C. The reaction time was 8 h. The crude product was purified by column chromatography using hexane/ethyl acetate (3:1) as the eluent to give enediyne **11b** (1.08 g, 73%) as a light brown solid. ^1^H NMR (400 MHz, CDCl_3,_ δ): 8.25–8.21 (m, 2H), 8.10–8.07 (m, 2H), 7.76–7.71 (m, 1H), 7.70–7.67 (m, 1H), 7.44–7.38 (m, 2H), 5.51 (t, *J* = 6.1 Hz, 1H), 4.46 (s, 2H), 3.50 (s, 3H), 3.40–3.35 (m, 2H), 2.73 (t, *J* = 6.1 Hz, 2H). ^13^C{^1^H} NMR (100 MHz, CDCl_3,_ δ): 150.0, 146.5, 138.6, 138.2, 128.3, 126.7, 125.8, 125.4, 124.4, 123.3, 123.0, 122.4, 95.3, 92.6, 79.6, 76.7, 60.8, 58.3, 42.2, 21.4. HRMS (ESI) *m*/*z*: [M+Na]^+^ Calcd for C_22_H_18_N_2_O_5_S_2_Na^+^, 477.0549; Found, 477.0557.

**4-Amino-*N*-(4-(2-(3-methoxyprop-1-yn-1-yl)benzo[*b*]thiophen-3-yl)but-3-yn-1-yl)benzenesulfonamide (11c).** To a stirred solution of enediyne **11b** (109 mg, 0.240 mmol, 1.00 equiv.) in DCM (3.20 mL) were added zinc dust (941 mg, 14.4 mmol, 60.0 equiv.). The reaction vial was cooled to 0 °C, and then glacial acetic acid (115 mg, 1.92 mmol, 0.110 mL, 8.00 equiv.) was added. The reaction mixture was allowed to warm to room temperature and was vigorously stirred at this temperature for 7.5 h (TLC control). After completion of the reaction, the reaction mixture was filtered through a pad of Celite; the sorbent was washed with ethyl acetate (75.0 mL). The resulting solution was washed two times with a saturated solution of NaHCO_3_ (2 × 75.0 mL) and two times with brine (2 × 75.0 mL), dried over anhydrous Na_2_SO_4_, and concentrated under reduced pressure to give the crude product, which was purified by column chromatography using benzene/acetonitrile (8:1) as the eluent to give NH_2_-enediyne **11c** (88.0 mg, 86%) as a light brown solid. ^1^H NMR (400 MHz, CD_3_CN, δ): 7.86–7.81 (m, 2H), 7.58–7.54 (m, 2H), 7.50–7.44 (m, 2H), 6.71–6.67 (m, 2H), 5.59 (t, *J* = 6.2 Hz, 1H), 4.79 (br s, 2H), 4.40 (s, 2H), 3.41 (s, 3H), 3.13–3.08 (m, 2H), 2.68 (t, *J* = 6.8 Hz, 2H). ^13^C{^1^H} NMR (100 MHz, CD_3_CN, δ): 153.2, 139.5, 139.2, 129.9, 127.9, 127.7, 126.5, 125.6, 124.3, 124.2, 123.4, 114.4, 96.6, 95.7, 79.5, 75.5, 60.9, 58.1, 42.9, 21.6. HRMS (ESI) *m*/*z*: [M+Na]^+^ Calcd for C_22_H_20_N_2_O_3_S_2_Na^+^, 447.0808; Found, 447.0815.

**Hexacarbonyl (*****N*-(4-(2-(3-methoxyprop-1-(1,2-η^2^)-yn-1-yl)benzo[*b*]thiophen-3-yl)but-3-yn-1-yl)-2-nitrobenzenesulfonamide)dicobalt (12a).** Cobalt complex **12a** was synthesized from enediyne **11a** (104 mg, 0.228 mmol, 1.00 equiv.) and octa-carbonyl dicobalt (82.0 mg, 0.239 mmol, 1.05 equiv) in absolute toluene (46.0 mL, *c* = 0.005 M) in accordance with the General Procedure (B). The reaction time was 1 h. Purification of the crude product by column chromatography using hexane/ethyl acetate (2:1) as the eluent gave complex **12a** (140 mg, 82%) as a dark red oil. ^1^H NMR (500 MHz, Acetone-*d_6_*, δ): 8.18 (br s, 1H), 7.90 (br s, 2H), 7.85 (br s, 3H), 7.45 (br s, 2H), 7.03 br (s, 1H), 4.94 (s, 2H), 3.59 (s, 3H), 3.55–3.51 (m, 2H), 2.93 (t, *J* = 6.6 Hz, 2H). ^13^C{^1^H} NMR (100 MHz, Acetone-*d_6_*, δ): 200.1, 149.1, 146.3, 141.7, 139.4, 134.9, 134.5, 133.6, 131.3, 126.9, 126.3, 125.8, 123.9, 123.2, 118.6, 98.0, 97.5, 77.8, 76.9, 73.9, 59.3, 43.2, 22.4. HRMS (ESI) *m*/*z*: [M+Na]^+^ Calcd for C_28_H_18_Co_2_N_2_O_11_S_2_Na^+^, 762.8908; Found, 762.8910.

**Hexacarbonyl (****4-amino-*N*-(4-(2-(3-methoxyprop-1-(1,2-η^2^)-yn-1-yl)benzo[*b*]thiophen-3-yl)but-3-yn-1-yl)benzenesulfonamide)dicobalt (12b).** Cobalt complex **12b** was synthesized from enediyne **11c** (30.0 mg, 0.707 mmol, 1.00 equiv.) and octa-carbonyl dicobalt (27.8 mg, 0.0813 mmol, 1.15 equiv.) in absolute toluene (14.0 mL, *c* = 0.005 M) in accordance with the General Procedure (B). The reaction time was 1 h. Purification of the crude product by column chromatography using hexane/ethyl acetate (3:2) as the eluent gave cobalt complex **12b** (44.4 mg, 88%) as dark violet solid. ^1^H NMR (400 MHz, CD_3_CN, δ): 7.82 (br s, 2H), 7.55 (d, *J* = 7.5 Hz, 2H), 7.45 (br s, 2H), 6.69 (d, *J* = 7.5 Hz, 2H), 5.57 (br s, 1H), 4.86 (s, 2H), 4.78 (br s, 2H), 3.53 (s, 3H), 3.17–3.05 (m, 2H), 2.68 (br s, 2H). ^13^C{^1^H} NMR (100 MHz, CD_3_CN, δ): 200.3, 153.6, 146.5, 141.7, 139.4, 129.9, 127.9, 127.0, 126.5, 123.9, 123.3, 114.4 (one of the «aromatic» signals overlaps with the CD_3_CN signal), 98.7, 97.2, 77.9, 76.7, 74.1, 59.4, 42.7, 22.1. HRMS (ESI) *m*/*z*: [M+Na]^+^ Calcd for C_28_H_20_Co_2_N_2_O_9_S_2_Na^+^, 732.9166; Found, 732.9169.

**Hexacarbonyl (*****N*-(4-(*N*-(4-(2-(3-methoxyprop-1-(1,2-η^2^)-yn-1-yl)benzo[*b*]thiophen-3-yl)but-3-yn-1-yl)sulfamoyl)phenyl)hept-6-ynamide)dicobalt (12c).** To a stirred solution of Co-complex **12b** (25.3 mg, 0.0356 mmol, 1.00 equiv.) in absolute THF (10.0 mL) under Ar, Et_3_N (5.41 mg, 0.0534 mmol, 7.42 μL, 1.50 equiv.) was added. The reaction mixture was cooled to 0 °C, and hex-5-ynoyl chloride (6.97 mg, 0.0534 mmol, 1.50 equiv.) was added. The reaction mixture was allowed to warm to room temperature and was stirred at room temperature for 1.5 h (TLC control). After completion of the reaction, the reaction mixture was poured into a saturated aqueous solution of NH_4_Cl (50,0 mL) and extracted with ethyl acetate (3 × 50.0 mL). The combined organic layers were washed two times with a 2% solution of NaOH (2 × 50.0 mL) and three times with brine (2 × 50.0 mL), dried over anhydrous Na_2_SO_4_, and concentrated under reduced pressure to give a crude product. The crude product was purified by column chromatography using hexane/acetone (2:1) as the eluent to give acylated Co-complex **12c** (16.7 mg, 58%) as a dark red oil. ^1^H NMR (400 MHz, Acetone-*d_6_*, δ): 9.48 (s, 1H), 7.91–7.81 (m, 6H), 7.48–7.44 (m, 2H), 6.75 (t, *J* = 6.1 Hz, 1H), 4.93 (s, 2H), 3.58 (s, 3H), 3.32–3.27 (m, 2H), 2.83 (t, *J* = 7.1 Hz, 2H), 2.55 (t, *J* = 7.4 Hz, 2H), 2.37 (t, *J* = 2.6 Hz, 1H), 2.29 (td, *J* = 7.0, 2.6 Hz, 2H), 1.92–1.85 (m, 2H). ^13^C{^1^H} NMR (100 MHz, Acetone-*d_6_*, δ): 200.1, 171.9, 146.2, 144.1, 141.7, 139.4, 135.8, 128.9, 126.8, 126.3, 124.0, 123.2, 119.7, 118.7, 98.4, 97.6, 84.3, 78.0, 76.7, 74.0, 70.5, 59.3, 42.8, 36.3, 24.9, 22.3, 18.3. Four signals (144.1, 135.8, 119.7, 42.8) double as a result of rotation around the amide bond. For the details, see the SI file. HRMS (ESI) *m*/*z*: [M+Na]^+^ Calcd for C_34_H_26_Co_2_N_2_O_10_S_2_Na^+^, 826.9585; Found, 826.9590.

**Hexacarbonyl ((1,2-η^2^)-****1,2,7,8-tetradehydro-4-((2-nitrophenyl)sulfonyl)-3,4,5,6-tetrahydrobenzo[4,5]thieno[2,3-*e*]azecine)dicobalt (13a).** Cyclic complex **13a** was synthesized in accordance with the general procedure (C) for the Nicholas reaction from Co-complex **12a** (44.0 mg, 0.0590 mmol, 1.00 equiv.) and boron trifluoride diethyl etherate (67.0 mg, 89.3 μL, 0.475 mmol, 8.00 equiv.) in absolute DCM (60.0 mL, *c* = 0.001 M). The reaction time was 1 h. Purification of the crude product by column chromatography using hexane/ethyl acetate (2:1) as the eluent gave cyclic complex **13a** (19.3 mg, 46%) as a dark brown oil. ^1^H NMR (400 MHz, Acetone-*d_6_*, δ): 8.20 (d, *J* = 7.7 Hz, 1H), 8.00–7.91 (m, 4H), 7.76 (d, *J* = 7.0 Hz, 1H), 7.50–7.44 (m, 2H), 5.34 (s, 2H), 3.89 (d, *J* = 4.8 Hz, 2H), 2.88 (d, *J* = 4.8 Hz, 2H). ^13^C{^1^H} NMR (100 MHz, Acetone-*d_6_*, δ): 199.9, 151.7, 150.1, 139.8, 139.4, 135.6, 133.1, 131.9, 131.1, 126.6 (two overlapping signals), 125.5, 123.6, 123.5, 116.8, 101.5, 98.6, 80.7, 79.8, 57.7, 55.5, 22.9. HRMS (ESI) *m*/*z*: [M+Na]^+^ Calcd for C_27_H_14_Co_2_N_2_O_10_S_2_Na^+^, 730.8646; Found, 730.8657.

**Hexacarbonyl (*N*-(4-(((1,2-η^2^)-1,2,7,8-tetradehydro-5,6-dihydrobenzo****[4,5]thieno[2,3-*e*]azecin-4(3*H*)-yl)sulfonyl)phenyl)hex-5-ynamide)dicobalt (13b).** Cyclic complex **13b** was synthesized in accordance with the general procedure (C) for the Nicholas reaction from Co-complex from complex **12c** (42.0 mg, 0.052.0 mmol, 1.00 equiv.) and boron trifluoride diethyl etherate (14.8 mg, 12.9 μL, 0.104 mmol, 2.00 equiv.) in absolute DCM (52.0 mL, *c* = 0.001 M). The reaction time was 1 h. Purification of the crude product by column chromatography using hexane/acetone (3:1) as the eluent gave cyclic complex **13b** (25.4 mg, 63%) as a dark red oil. ^1^H NMR (400 MHz, Acetone-*d_6_*, δ): 9.59 (br s, 1H), 7.96–7.90 (m, 5H), 7.76–7.74 (m, 1H), 7.49–7.39 (m, 2H), 4.97 (s, 2H), 3.74 (d, *J* = 5.3 Hz, 2H), 2.84 (d, *J* = 5.3 Hz, 2H), 2.57 (t, *J* = 7.3 Hz, 2H), 2.38 (t, *J* = 2.6 Hz, 1H), 2.30 (td, *J* = 7.0, 2.6 Hz, 2H), 1.93–1.86 (m, 2H). ^13^C{^1^H} NMR (100 MHz, Acetone-*d_6_*, δ): 200.1, 172.1, 151.5, 144.8, 139.8, 139.4, 132.8, 129, 126.5, 123.6, 123.5, 119.9, 116.8, 101.4, 99.1, 84.3, 80.6, 79.4, 70.5, 57.8, 56.4, 36.3, 24.8, 23.6, 18.3 (two «aromatic» CH signals overlap). Three signals (144.8, 119.9, 36.3) double as a result of rotation around the amide bond. For the details, see the SI file. HRMS (ESI) *m*/*z*: [M+Na]^+^ Calcd for C_33_H_22_Co_2_N_2_O_9_S_2_Na^+^, 794.9323; Found, 794.9311.

**1,2,7,8-tetradehydro-4-((2-nitrophenyl)sulfonyl)-3,4,5,6-tetrahydrobenzo[4,5]thieno[2,3-*e*]azecine (II).** Enediyne **II** was synthesized in accordance with the General Procedure (D) for the Co-complexes deprotection from complex **13a** (33.0 mg, 0.0466 mmol, 1.00 equiv.) and TBAF trihydrate (557 mg, 1.77 mmol, 65.8 equiv.) in a mixture of acetone/water (15:1, *v/v*, 4.50 mL, c = 0.006 M). TBAF was added in ten portions with an interval of 30 min. The reaction time was 4.5 h. The crude product was purified by column chromatography using hexane/acetone (3:1) as the eluent to give enediyne II (7.30 mg, 64%) as an orange solid. ^1^H NMR (400 MHz, Acetone-*d_6_*, δ): 8.23 (d, *J* = 8.0 Hz, 2H), 7.99–7.79 (m, 5H), 7.53–7.47 (m, 2H), 4.63 (s, 2H), 3.91 (d, *J* = 5.1 Hz, 2H), 3.01 (d, *J* = 5.1 Hz, 2H). ^13^C{^1^H} NMR (100 MHz, Acetone-*d_6_*, δ): 139.0, 136.5, 135.6, 133.1, 132.1, 131.3, 130.5, 130.4, 129.0, 127.3, 126.5, 125.4, 124.0, 123.5, 104.5, 102.1, 82.9, 79.5, 52.4, 43.7, 22.9. HRMS (ESI) *m*/*z*: [M+H]^+^ Calcd for C_21_H_15_N_2_O_4_S_2_^+^, 423.0468; Found, 423.0469.

***N*-(4-((1,2,7,8-tetradehydro-5,6-dihydrobenzo****[4,5]thieno[2,3-*e*]azecin-4(3*H*)-yl)sulfonyl)phenyl)hex-5-ynamide (III).** Enediyne **III** was synthesized in accordance with the General Procedure (D) for the Co-complexes deprotection from complex **13b** (22.0 mg, 0.0285 mmol, 1.00 equiv.) TBAF trihydrate (370 mg, 1.17 mmol, 41.0 equiv.) in a mixture of acetone/water (15:1, *v/v*, 4.80 mL, c = 0.006 M). TBAF trihydrate was added in eight portions with an interval of 40 min. The reaction time was 5.5 h. The crude product was purified by column chromatography using hexane/acetone (2:1) as the eluent to give enediyne **III** (10.5 mg, 77%) as a light red solid. ^1^H NMR (400 MHz, Acetone-*d_6_*, δ): 9.52 (br s, 1H), 7.98–7.94 (m, 1H), 7.90–7.85 (m, 4H), 7.80–7.76 (m, 1H), 7.52–7.45 (m, 2H), 4.38 (s, 2H), 3.64 (d, *J* = 5.0 Hz, 2H), 2.96 (d, *J* = 5.0 Hz, 2H), 2.54 (t, *J* = 7.4 Hz, 2H), 2.37 (t, *J* = 2.6 Hz, 1H), 2.28 (td, *J* = 7.0, 2.6 Hz, 2H), 1.91–1.83 (m, 2H). ^13^C{^1^H} NMR (100 MHz, Acetone-*d_6_*, δ): 172.0, 144.7, 138.9, 136.6, 133.0, 130.3, 129.4, 129.0, 127.2, 126.4, 124.0, 123.4, 119.8, 105.0, 102.3, 84.3, 82.5, 79.3, 70.5, 52.3, 43.3, 36.3, 24.8, 23.4, 18.3. Three signals (144.7, 119.8, 36.3) double as a result of rotation around the amide bond. For the details, see the SI file. HRMS (ESI) *m*/*z*: [M+H]^+^ Calcd for C_27_H_23_N_2_O_3_S_2_^+^, 487.1145; Found, 487.1142.

**1,2,7,8-tetradehydro-4-(phenylsulfonyl)-3,4,5,6-tetrahydrobenzo****[4,5]thieno[2,3-*e*]azecine (14).** Enediyne **14** was synthesized in accordance with the General Procedure (D) for the Co-complexes deprotection from Co-complex **13c** (120 mg, 0.177 mmol, 1.00 equiv.) and TBAF trihydrate (1.95 g, 6.195 mmol, 35.0 equiv.) in an acetone/water mixture (15:1, *v/v*, 25.0 mL, *c* = 0.006 M). TBAF trihydrate was added in eight portions with an interval of 0.5 h. The reaction time was 3.5 h. The crude product was purified by column chromatography using hexane/acetone (3:1) as the eluent to give NTs-enediyne **14** (52.1 mg, 75%) as a beige solid. ^1^H NMR (400 MHz, Acetone-*d_6_*) δ 7.98–7.96 (m, 1H), 7.83 (d, *J* = 8.3 Hz, 2H), 7.79–7.77 (m, 1H), 7.53–7.46 (m, 2H), 7.44 (d, *J* = 8.3 Hz, 2H), 4.39 (s, 2H), 3.64 (d, *J* = 5.0 Hz, 2H), 2.97 (d, *J* = 5.0 Hz, 2H), 2.41 (s, 3H).^1^H NMR spectrum corresponds with the data reported earlier [42].

***N*-benzyl-4-(2-(3-methoxyprop-1-yn-1-yl)benzo[*b*]thiophen-3-yl)but-3-yn-1-amine (16a).** Enediyne **16a** was synthesized in accordance with the General Procedure (A) for the Sonogashira coupling from 3-iodobenzothiophene **1** (328 mg, 1.00 mmol, 1.00 equiv.), *N*-benzylbut-3-yn-1-amine (**15a**) (249 mg, 1.50 mmol, 1.50 equiv.), KF (464 mg, 8.00 mmol, 8.00 equiv.), Pd(PPh_3_)_4_ (58.0 mg, 0.0500 mmol, 5 mol%) and CuI (28.0 mg, 0.150 mmol, 15 mol%) in DMF (10.0 mL) at 40 °C. The reaction time was 6 h. Purification of the crude product by column chromatography using hexane/ethyl acetate (2:1→1:1) as the eluent gave enediyne **16a** (185 mg, 52%) as an orange oil. ^1^H NMR (400 MHz, Acetone-*d_6_*, δ): 7.94–7.90 (m, 1H), 7.89–7.84 (m, 1H), 7.51–7.47 (m, 2H), 7.41 (d, *J* = 7.1 Hz, 2H), 7.33–7.29 (m, 2H), 7.23 (t, *J* = 7.3 Hz, 1H), 4.37 (s, 2H), 3.88 (s, 2H), 3.40 (s, 3H), 2.94 (t, *J* = 6.7 Hz, 2H), 2.78 (t, *J* = 6.7 Hz, 2H; br s, 1H–NH, the signal overlaps with the water signal). ^13^C{^1^H} NMR (100 MHz, Acetone-*d_6_*, δ): 141.8, 139.5, 139.0, 129.0, 128.9, 127.52, 127.51, 126.3, 125.1, 124.5, 124.1, 123.3, 97.3, 96.3, 79.5, 74.9, 60.6, 57.7, 53.8, 48.6, 21.4. HRMS (ESI) *m*/*z*: [M+H]^+^ Calcd for C_23_H_21_NOS^+^ 360.1417; Found, 360.1406. 

***N*-(4-(2-(3-methoxyprop-1-yn-1-yl)benzo[*b*]thiophen-3-yl)but-3-yn-1-yl)benzamide (16b).** Enediyne **16b** was synthesized in accordance with the General Procedure (A) for the Sonogashira coupling from 3-iodobenzothiophene **1** (100 mg, 0.303 mmol, 1.00 equiv.), *N*-(but-3-yn-1-yl)benzamide (**15b**) (68.0 mg, 0.394 mmol, 1.30 equiv), KF (88.3 mg, 1.52 mmol, 5.00 equiv.), Pd(PPh_3_)_4_ (17.5 mg, 0.0152 mmol, 5 mol%) and CuI (8.6 mg, 0.0456 mmol, 15 mol%) in DMF (5.00 mL) at 50 °C. The reaction time was 5 h. Purification of the crude product by column chromatography using hexane/ethyl acetate (2:1) as the eluent gave enediyne **16a** (92.0 mg, 81%) as a yellow oil. ^1^H NMR (400 MHz, CDCl_3_, δ):7.84−7.80 (m, 2H), 7.72 (d, *J* = 7.1 Hz, 1H), 7.51–7.35 (m, 5H), 6.81 (br. S, 1H), 4.29 (s, 2H), 7.79–7.75 (m, 2H), 3.39 (s, 3H), 2.91 (t, *J* = 6.2 Hz, 2H). ^13^C{^1^H} NMR (101 MHz, CDCl_3_, δ): 167.8, 138.0, 138.5, 134.6, 131.7, 128.7, 127.1, 126.5, 125.3, 125.21, 123.49, 123.4, 122.3, 94.9, 94.5, 79.6, 75.6, 60.6, 58.0, 39.0, 21.0. HRMS (ESI) *m*/*z*: [M+Na]^+^ Calcd for C_23_H_19_NO_2_Sna^+^, 396.1029; Found, 396.1030.

**Hexacarbonyl (*****N*-benzyl-4-(2-(3-methoxyprop-1-(1,2-η^2^)-yn-1-yl)benzo[*b*]thiophen-3-yl)but-3-yn-1-amine)dicobalt (17a).** Cobalt complex **17a** was synthesized from enediyne **16a** (89.9 mg, 0.250 mmol, 1.00 equiv.) and octa-carbonyl dicobalt (94.0 mg, 0.275 mmol, 1.1 equiv.) in absolute toluene (50.0 mL, *c* = 0.005 M) in accordance with the General Procedure (B). The reaction time was 1 h. Purification of the crude product by column chromatography using hexane/ethyl acetate (3:1) as the eluent gave cobalt complex **17a** (111 mg, 69%) as a dark red-brown solid. ^1^H NMR (400 MHz, Acetone-*d_6_*, δ): 7.90–7.84 (m, 2H), 7.46–7.39 (m, 4H), 7.33–7.29 (m, 2H), 7.26–7.21 (m, 1H), 4.96 (s, 2H), 3.88 (s, 2H), 3.57 (s, 3H), 2.98 (br s, 2H), 2.86–2.78 (m, 2H, NH, overlaps with the water signal). ^13^C{^1^H} NMR (101 MHz, Acetone-*d_6_*, δ): 200.1, 145.8, 141.9, 141.8, 139.5, 129.0, 128.9, 127.5, 126.8, 126.3, 123.8, 123.2, 119.0, 100.4, 97.6, 78.2, 76.2, 74.0, 59.2, 54.0, 48.6, 22.0. HRMS (ESI) *m*/*z*: [M+H]^+^ Calcd for C_29_H_22_Co_2_NO_7_S^+^, 645.9775; Found, 645.9806.

**Hexacarbonyl (*N*-(4-(2-(3-methoxyprop-1****-(1,2-η^2^)-yn-1-yl)benzo[*b*]thiophen-3-yl)but-3-yn-1-yl)benzamide)dicobalt (17b).** Cobalt complex **17b** was synthesized from enediyne **16b** (70.0 mg, 0.186 mmol, 1.00 equiv.) and octa-carbonyl dicobalt (70.0 mg, 0.205 mmol, 1.10 equiv.) in absolute toluene (37.2 mL, *c* = 0.005 M) in accordance with the General Procedure (B). The reaction time was 1 h. Purification of the crude product by column chromatography using hexane/ethyl acetate (2:1) as the eluent gave cobalt complex **17b** (113 mg, 92%) as a burgundy oil. ^1^H NMR (400 MHz, CDCl_3_, δ): 7.83–7.72 (m, 4H), 7.53–7.33 (m, 5H), 6.58 (br s, 1H), 4.87 (s, 2H), 3.81–3.77 (m, 2H), 3.59 (s, 3H), 2.91 (t, *J* = 6.5 Hz, 2H). Only signals from CO ligands and carbon atoms bonded to hydrogen atoms can be detected by ^13^C{^1^H} NMR. ^13^C{^1^H} NMR (125 MHs, CDCl_3_, δ): 199.2, 132.0, 129.0, 127.3, 126.1, 125.5, 123.1, 122.4, 73.7, 59.6, 39.1, 21.5. HRMS (ESI) *m*/*z*: [M+Na]^+^ Calcd for C_29_H_19_Co_2_NO_8_SNa^+^, 681.9388; Found, 681.9389.

**Hexacarbonyl****(4-benzyl-(1,2-η^2^)-1,2,7,8-tetradehydro-3,4,5,6-tetrahydrobenzo[4,5]thieno[2,3-*e*]azecine)dicobalt (18). A)** Cyclic Co-complex **18** was synthesized in accordance with the general procedure (C) for the Nicholas reaction from Co-complex **17a** (30.0 mg, 0.0465 mmol, 1.00 equiv.) and boron trifluoride diethyl etherate (29.7 mg, 25.8 μL, 0.209 mmol, 4.50 equiv.) in absolute DCM (46.5 mL, *c* = 0.001 M). The reaction time was 20 h. Purification of the crude product by column chromatography using hexane/ethyl acetate (30:1) as the eluent gave cyclic complex **18** (1.50 mg, 4%) as a dark red-brown solid.

**(B)** Cyclic Co-complex **18** was synthesized in accordance with the general procedure (C) for the Nicholas reaction from Co-complex **17a** (73.4 mg, 0.114 mmol, 1.00 equiv.) and another acid–tetrafluoroboric acid diethyl ether complex (66.7 mg, 55.8 μL, 0.410 mmol, 3.60 equiv.) in absolute DCM (114 mL, *c* = 0.001 M). The reaction time was 20 h. Purification of the crude product by column chromatography using hexane/ethyl acetate (30:1) as the eluent gave cyclic complex **18** (3.90 mg, 6%) as a dark red-brown solid.

**(C)** To an argon-flushed, cooled (–20 °C) stirred solution of Co_2_(CO)_6_-complex of an acyclic enediyne **17a** (44.0 mg, 0.0682 mmol, 1.00 equiv.) in anhydrous DCM (44.0 mL, *c* = 0.001 M) was added tetrafluoroboric acid diethyl ether complex (33.1 mg, 28.1 μL, 0.205 mmol, 3.00 equiv.). The resulting mixture was stirred at –20 °C for 10 min; then *N,N*-diisopropylethylamine (DIPEA) (26.4 mg, 35.6 μL, 0.205 mmol, 3.00 equiv.) was added. The reaction mixture was allowed to warm to room temperature and was stirred at this temperature for 1 h. TLC analysis showed the presence of traces of the product **18**. Therefore, the reaction mixture was cooled to –20 °C and treated with an additional amount of HBF_4_ × Et_2_O (33.1 mg, 28.1 μL, 0.205 mmol, 3.00 equiv.) followed by the additional amount of DIPEA (26.4 mg, 35.6 μL, 0.205 mmol, 3.00 equiv.). The reaction mixture was allowed to warm to room temperature and was stirred at this temperature for 3 h. Then it was cooled to –20 °C once again and treated with HBF_4_ × Et_2_O (22.1 mg, 18.7 μL, 0.136 mmol, 2.00 equiv.) and after 10 min additionally with DIPEA (26.4 mg, 35.6 μL, 0.205 mmol, 3.00 equiv.). The reaction mixture was allowed to warm to room temperature and was stirred at this temperature for 3 h. Then the reaction mixture was quenched with a saturated aqueous solution of NaHCO_3_. The organic layer was separated, washed with brine, dried over anhydrous Na_2_SO_4_, and concentrated under reduced pressure to yield a crude product, which was purified by column chromatography using hexane/ethyl acetate (30:1) as the eluent gave cyclic complex **18** (4.90 mg, 12%) as a dark red-brown solid. ^1^H NMR (400 MHz, Acetone-*d_6_*, δ): 7.96–7.92 (m, 1H), 7.81–7.78 (m, 1H), 7.50–7.44 (m, 4H), 7.39–7.33 (m, 2H), 7.29–7.26 (m, 1H), 4.57 (s, 2H), 4.17 (s, 2H), 3.15 (t, *J* = 5.8 Hz, 2H), 2.55 (t, *J* = 5.8 Hz, 2H). ^13^C{^1^H} NMR (100 MHz, Acetone-*d_6_*, δ): 200.5, 150.2, 140.2, 139.8, 139.5, 129.5, 129.3, 128.0, 126.3, 126.5, 123.8, 123.6, 118.0, 102.1, 101.6, 81.2, 79.5, 61.6, 61.1, 55.5, 22.4. HRMS (ESI) *m*/*z*: [M+H]^+^ Calcd for C_28_H_18_Co_2_NO_6_S^+^ 613.9513; Found, 613.9531.

**(5-(2-(3-Methoxyprop-(1,2-η^2^)-1-yn-1-yl)benzo[*b*]thiophen-3-yl)-2,3-dihydro-1*H*-pyrrol-1-yl)(phenyl)methanone (19).** Co-complex of pyrroline derivative **19** was synthesized in accordance with the general procedure (C) for the Nicholas reaction from Co-complex **17b** (52.0 mg, 0.079 mmol, 1.00 equiv.) and boron trifluoride diethyl etherate (14.5 mg, 13.0 μL, 0.102 mmol, 1.30 equiv.) in absolute DCM (79.0 mL, *c* = 0.001 M). The reaction time was 2 h. Purification of the crude product by column chromatography using hexane/ethyl acetate (10:1) as the eluent gave Co-complex of pyrroline derivative **19** (22.0 mg, 44%) as a burgundy solid. ^1^H NMR (400 MHz, CDCl_3_, δ) 7.51 (br s, 1H), 7.44 (br s, 1H), 7.29–7.12 (m, 4H), 7.12–6.87 (m, 3H), 5.46 (br s, 1H), 4.74 (d, *J* = 12.9 Hz, 1H), 4.70 (d, *J* = 12.9 Hz, 1H), 4.38–4.31 (m, 1H), 4.19 (br s, 1H), 3.61 (s, 3H), 2.92–2.75 (m, 2H). NMR DEPT (101 MHz, CDCl_3_, δ): 129.7 (CH), 126.9 (CH–two signals overlap), 124.9 (CH), 124.6 (CH), 122.4 (CH), 121.6 (CH), 117.1 (CH), 73.6 (OCH_2_), 59.2 (OCH_3_), 50.1 (CH_2_), 28.0 (OCH_2_). The ^13^C NMR was too broad and had a low signal-to-noise ratio. For the details, see the Appendix A. HRMS (ESI) *m*/*z*: [M+Na]^+^ Calcd for C_29_H_19_Co_2_NO_8_SNa^+^, 681.9388; Found, 681.9412.

### 4.4. Cell Culture

NCI-H460 lung carcinoma cells and WI-26 VA4 lung epithelial-like cells were purchased from the ATCC. NCI-H460 cells were maintained in Advanced RPMI-1640 (Gibco, UK) supplemented with 5% fetal bovine serum (FBS, Gibco, Leicestershire, UK), penicillin (100 UI mL^−1^), streptomycin (100 µg mL^−1^) and GlutaMax (2 mM, Gibco, UK). WI-26 VA4 cells were maintained in Advanced MEM (Gibco, UK) supplemented with 5% fetal bovine serum (FBS, Gibco, UK), penicillin (100 UI mL^−1^), streptomycin (100 µg mL^−1^), and GlutaMax (1.87 mM, Gibco, UK). All cells line cultivation under a humidified atmosphere of 95% air/5% CO_2_ at 37 °C. Subconfluent monolayers, in the log growth phase, were harvested by a brief treatment with TrypLE Express solution (Gibco, UK) in phosphate-buffered saline (PBS, Capricorn Scientific, Ebsdorfergrund, Germany) and washed three times in serum-free PBS. The number of viable cells was determined by trypan blue exclusion.

### 4.5. Antiproliferative Assay

The effects of the synthesized compounds on cell viability were determined using the MTT colorimetric test. All examined cells were diluted with the growth medium to 3.5 × 104 cells per mL, and the aliquots (7 × 103 cells per 200 μL) were placed in individual wells in 96-multiplates (Eppendorf, Germany) and incubated for 24 h. The next day the cells were then treated with synthesized compounds separately at the final concentration of 75 μM and incubated for 72 h at 37 °C in a 5% CO_2_ atmosphere. After incubation, the cells were then treated with 40 μL MTT solution (3-(4,5-dimethylthiazol-2-yl)-2,5-diphenyltetrazolium bromide, 5 mg mL^−1^ in PBS) and incubated for 4 h. After an additional 4 h incubation, the medium with MTT was removed, and DMSO (150 μL) was added to dissolve the crystals formazan. The plates were shaken for 10 min. The optical density of each well was determined at 560 nm using a microplate reader GloMax Multi+ (Promega, Madison, WI, USA). Each of the tested compounds was evaluated for cytotoxicity in three separate experiments.

## 5. Conclusions

The scope and limitation of the Nicholas-type cyclization for the synthesis of various 10-membered heteroenediynes fused to a benzothiophene ring were studied. We proved that an arenesulfonamide fragment is the optimal functional group to realize a high-yield synthesis of 10-membered enediynes. Moreover, this group can serve as a site for the introduction of additional functional groups for further modification of cyclic enediynes via click chemistry. For this purpose, the 4-(*N*-acylamino)benzenesulfonamide functional group can be used as a nucleophile for cyclization and functionalization. The crucial point is that neither the secondary amino group nor the amido functional group is suitable for the closure of the 10-membered azaendiyne, which is a limitation of the aza-Nicholas cyclization. Functionalized O-enediynes are also synthetically accessible through the O-Nicholas reaction. Still, the use of such functionalization is limited because of the low stability of O-enediynes compared with that of N-enediynes. All the synthesized cyclic enediynes were tested as potential anticancer compounds and showed moderate activity against NCI-H460 lung carcinoma and had a minimal effect on WI-26 VA4 lung epithelial-like cells, demonstrating that the synthesized enediynes can be further used to synthesize active molecules with antitumor activity based on enediyne conjugates. Azaenediynes modified through acylated 4-aminobenzenesulfonamide nucleophilic group should be considered the most suitable structures for ongoing studies.

## Data Availability

Data supporting reported results (copies of ^1^H, ^13^C{^1^H}, DEPT, 2D NMR spectra in PDF) can be found in a Supporting Information File or can be sent as original files upon request.

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
