# Peer review of "Functionalized 10-Membered Aza- and Oxaenediynes through the Nicholas Reaction"

_molecules, 2022, doi:10.3390/molecules27186071_

Round 1

Reviewer 1 Report

Many scientists are starting to turn their attention to ten-membered enediynes because the DNA-damaging effects of calicheamicin result in apoptosis and cell death at remarkably low concentrations.

The authors presented outstanding research on 10-membered heteroenediynes. 3-Iodo-2-(3-methoxyprop-1-yn-1-yl)benzo[b]thiophene was used as starting reagent, Sonogashira coupling led to unsymmetrically substituted acyclic enediynes with the required functionalities at both triple bonds. Complexes Co2(CO)6 were formed, and Nicholas cyclization of these unique molecules was studied. Different stability of O-enediynes was demonstrated. Bergmann cyclization of oxa-enediynes was carried out in isopropanol at slight heating. N-enediyne molecules were constructed, competitive processes of pyrroline ring and 10-membered enediyne formation are considered, limitations of the Nicholas reaction are discussed.

Structures of target products and intermediates were confirmed by 1H and 13C NMR spectra, DEPT experiments were used. Moderate activity of synthesized heteroendiynes against NCI-H460 lung carcinoma was demonstrated.

The manuscript can be accepted after minor revision. Some comments:

Scheme 2 – TBAF instead of TFAF

Page 5 – abbreviation Ns-group, please write the full name at the first mention

Supplementary – Copies of MS spectra for key compounds should be included. Separate chapter with several mass-spectra would be great in addition to the picture for compound 24 in the page S89.

Author Response

We are grateful to  the Reviewer for the evaluating of our manuscript and valubal comments. All comments have been carefully studied and taken into account in the processing of the revised version of manuscript.

The reviewers’s questions (highlighted in red text) with the answers are given below in italics:

Scheme 2 – TBAF instead of TFAF - done

Page 5 – abbreviation Ns-group, please write the full name at the first mention – done, and the same for Ts

Supplementary – Copies of MS spectra for key compounds should be included. Separate chapter with several mass-spectra would be great in addition to the picture for compound 24 in the page S89.

The SI was corrected: spectra for compounds which are not in the manuscript were removed and MS of all cyclic enediynes and Co-complexes 18 and 19 were added.

Reviewer 2 Report

The manuscript entitled "Functionalized 10-Membered Aza- and Oxa-Enediynes through the Nicholas Reaction" by Balova et al. reports the titular topic with broad substrate range and significant advancement of present status in this area. The compounds are well characterized by NMR, DEPT NMR, etc. Therefore, I suggest acceptance of this manuscript after minor revisions on following points.

1. The manuscript needs to be corrected with some typo. e.g., Scheme 2 "TFAF"

2. Buchwald-Hartwig cross-coupling? KF is quite a strong base. It is expected to provide C-N coupling also. Do authors have any suggestive idea how their synthons did not react by this coupling at all?

3. Also the Glaser coupling? Is it not possible in the Sonogashira cross-coupling reactions? e.g., Scheme 3 step-I, what are the side-products? why there is so much yield variation ? The authors need to highlight these areas with some possible explanation and citations.

a recent reference (on cross-Sonogashira on NH2 containing system) to be added here: ACS Omega 2022, 7, 29, 25874–25880 

Author Response

We are grateful to the Reviewer for the evaluating of our Manuscript.  All comments have been carefully studied and taken into account in the processing of the manuscript. The reviewer’s questions with the answers (highlighted in red text) are given in the attaced file.
